# Evaluating language policy implementation in South African higher education - three decades of progress and challenges: A scoping review protocol

Silingene Joyce Ngcobo[1‡], Tracy Zhandire[1*‡], Zamasomi Meyiwa Luvuno[2], Wilbroda Hlolisile Chiya[1], Celenkosini Thembelenkosini Nxumalo[1], Gugulethu Brightness Mazibuko[3], Busisiwe Purity Ncama[1], Sinegugu Evidence Duma[1], Deshini Naidoo[4]

1 College of Health Sciences, School of Nursing and Public Health, Howard College Campus, University of KwaZuluNatal, Durban, South Africa, 2 Centre for Rural Health, School of Nursing and Pubic Health, Howard Campus College, University of KwaZulu-Natal, Durban, South Africa, 3 College of Humanities, School of Art, Howard College Campus, University of KwaZulu--Natal, Durban, South Africa, 4 College of Health Sciences, School of Health Sciences, Westville Campus, University of KwaZulu-Natal, Durban, South Africa

☯ These authors contributed equally to this work.
‡ SJN and TZ also contributed equally to this work.
* tzhandire@gmail.com

## Abstract

### Background

South Africa's higher education institutions (HEIs) continue to face challenges in implementing inclusive language policies that integrate indigenous African languages into academic settings, even three decades after apartheid. Higher Education Institutions (HEIs) face significant challenges in integrating indigenous African languages into academic settings. Despite progressive reforms, higher education institutions face significant challenges in integrating indigenous African languages into academic settings.

### Objectives

This scoping review aims to evaluate the current state of language policy implementation in South African public HEis. Specifically, it seeks to: (1) map the integration of multilingual policies into teaching, research, and administrative practices; (2) identify persistent barriers to effective policy implementation; (3) explore successful strategies for promoting multilingualism (4) assess the extent of African language usage in academic contexts; and (5) identify research gaps to guide future investigations.

### Methods

The review will adhere to the PRISMA-ScR guidelines and follow the framework outlined by Arksey and O'Malley, ensuring a systematic and transparent approach.

**Data availability statement:** No data are currently available as this protocol outlines the planned methodology for a scoping review. Upon completion, all relevant datasets will be deposited in the Open Science Framework (OSF) repository and made publicly accessible at https://doi.org/10.17605/OSF.IO/AU2SD

**Funding:** The author(s) received no specific funding for this work.

**Competing interests:** The authors have declared that no competing interest exist.

A comprehensive search will be conducted in databases including Google Scholar, Scopus, Web of Science, ERIC, and African Journals Online (AJOL), covering studies published from 1994 to the present. This will be supplemented by grey literature from government and institutional sources. Three independent reviewers will screen studies using predefined eligibility criteria, managing and screening articles through Rayyan. Data will be extracted using a standardized form, and thematic analysis will synthesize the findings, with stakeholder consultation to validate results.

## Expected outcomes

This review will provide a comprehensive assessment of language policy implementation, highlighting successful strategies and persistent challenges across institutions. The findings will inform policy refinement, identify effective practices, and guide future research directions for achieving linguistically inclusive higher education in South Africa, while contributing to a broader understanding of implementing multilingual policies in post-colonial educational contexts.

This protocol is preregistered on OSF, available at https://doi.org/10.17605/OSF.IO/AU2SD

## Background

The importance of language in education, particularly in African contexts, cannot be overstated. One of the primary learning challenges for African children is linguistic barriers [1]. This highlights the urgent need for policymakers and education sector donors to strengthen African languages as mediums of instruction, especially in foundational education [2]. Xulu-Gama and Hadebe [3], argues for comprehensive language policies spanning all educational levels, including higher education, to enhance accessibility and inclusivity.

South African higher education institutions (HEIs), which encompass public universities, technical universities, and vocational colleges, face ongoing challenges in implementing language policies that reflect the country's linguistic diversity and address historical inequities [4,5]. Post-apartheid language reforms, however, have made significant strides. The 1996 Constitution recognized 11 official languages and mandated multilingualism across all sectors, including education [6]. Following this, the 2002 Language Policy for Higher Education emphasized promoting African languages alongside English and Afrikaans to redress historical injustices and foster linguistic equity in HEIs [7].

Further, the 2020 Language Policy Framework for Public Higher Education Institutions reaffirmed these commitments, requiring institutions to revise their policies and prioritize historically marginalized languages for academic purposes [8]. In 2023, the South African Sign Language Bill recognized South African Sign Language (SASL) as the 12th official language, extending inclusivity to the Deaf community [9].

Despite progressive policies, persistent challenges remain. These include resource constraints, insufficient academic resources in African languages, a shortage of trained staff, and the continued dominance of English in academia [10,11].

While some institutions, such as the University of KwaZulu-Natal, have successfully integrated indigenous languages into their teaching and administrative practices [12], others struggle with effective implementation [13].

As South Africa approaches three decades of democratic education, it is imperative to systematically evaluate language policy implementation in HEIs. This will help identify successful strategies, address barriers, and refine policies to better promote multilingualism.

### Study rationale

Language is a critical factor in promoting equity and inclusivity in higher education. Despite comprehensive policies designed to integrate African languages into teaching, research, and administration, implementation has been uneven. Challenges such as limited resources, inadequate academic materials, and the predominance of English continue to hinder multilingualism. Evaluating the current implementation status of language policies is essential for addressing these gaps and identifying effective strategies. This scoping review will contribute to policy refinement, highlight effective practices, and offer actionable insights for fostering linguistic equity in South Africa's HEIs.

### Aim

To examine the current implementation status of language policies in South African public higher education institutions and evaluate their effectiveness nearly three decades after the introduction of post-apartheid language reforms.

### Objectives

The main question will be addressed through the following research objectives:

1. To map the integration of multilingual policies into institutional teaching, research, and administrative practices.

2. To identify barriers to the effective implementation of language policies.

3. To explore successful strategies employed by institutions to promote multilingualism.

4. To evaluate the extent of multilingualism, with a particular focus on the use of African languages in HEIs.

5. To identify research gaps that can guide future investigations and policy interventions.

## Methodology

### Scoping review framework

This scoping review will follow the framework outlined by Arksey and O'Malley [14], which has been refined by Levac, Colquhoun [15] to ensure rigor and transparency. The review will be conducted with guidance from the 2020 version of the JBI Manual for Evidence Synthesis [16] and will adhere to the PRISMA-ScR guidelines [17]. The review will be organized according to Arksey's six stages: (1) identifying the research question; (2) identifying relevant studies; (3) study selection; (4) charting the data; (5) collating, summarizing and reporting the results; and (6) consultation to inform and validate findings.

### Stage 1: Identifying the research question

Collaborative team discussions and a preliminary literature review have defined the research question: *What is the current implementation status of language policy in South African public higher education institutions?* This question aligns with the study's objectives, as detailed in the background section, to explore the integration of multilingual policies, barriers, strategies, and gaps in research.

**Eligibility of research questions.** The inclusion criteria for studies in this review will be developed using the Population, Concept, Context (PCC) framework [18]. This framework will guide the determination of study eligibility, aligning with the research question as outlined in Table 1. The PCC framework ensures that the studies selected are relevant to the population of interest, the concepts being explored, and the context in which the research is conducted.

## Stage 2: Identifying relevant studies

**Search strategy.** A comprehensive search strategy will be employed to identify literature from Google Scholar, Scopus, Web of Science, ERIC, and African Journals Online (AJOL). The search will cover studies published from 1994 to the present, aligning with South Africa's post-apartheid era. This period was marked by significant language policy reforms, including the adoption of the 1996 Constitution and subsequent multilingual education policies. Restricting the search to this timeframe ensures that the review captures literature reflecting these policy shifts and their impact on higher education over the past three decades.

**Search strategy adaptation for different databases.** To ensure a comprehensive and systematic literature search, multiple search strings have been developed and will be adapted based on the capabilities of each database. The search strategy will incorporate Boolean operators (AND, OR), truncation (*), and proximity searching (NEAR/N) where applicable:

1. Scopus, Web of Science, and ERIC, proximity operators (NEAR/N) and Boolean logic will be used to refine searches and improve accuracy.

2. Google Scholar, a simplified Boolean approach will be applied due to platform limitations.

3. African Journals Online (AJOL)**, t**he search will focus on key terms while considering character limits and indexing constraints.

By using multiple databases, this strategy ensures broad coverage, capturing both peer-reviewed studies and grey literature relevant to language policy implementation in South African universities.

Table 2 presents the search string developed for databases Web of Science, Scopus, and ERIC. The string has been designed to capture comprehensive results related to language policy, governance, and implementation in higher education contexts, specifically focusing on South African universities. The search terms include a combination of synonyms and related keywords for key concepts, such as "language policy" and "higher education;" as well as a broad range of official

Table 1. PCC table to determine eligibility of research questions.

| PCC Element | Components | Definition |
|---|---|---|
| **Population** | Institution type | Public higher education institutions (HEIs) in South Africa. |
| | Stakeholder | Academic staff, administrative staff, students (undergraduates and postgraduates), and language policy committees/units. |
| **Concept** | Implementation status | Current implementation of language policies and multilingual approaches. |
| | Key areas | Teaching, learning, research, administrative functions, barriers, and successful strategies for multilingual promotion. |
| **Context** | Time period | Post-apartheid era (1994 - present) |
| | Policy Framework | National and institutional language policies and the higher education transformation agenda. |
| | Source Types | Peer-reviewed literature, grey literature, government reports, institutional documentation. |

**Table 2. Search string 1 for web of scince, scopus and ERIC databases (focused on South African language policy in higher education.**

| |
|---|
| ((ALL("language policy") NEAR/2 (governance OR planning OR framework OR implementation OR enforcement)) |
| AND (ALL("higher education") NEAR/3 (South Africa OR university OR "tertiary education" OR "postsecondary education")) |
| AND (ALL(multilingual* OR bilingual* OR "language diversity" OR "indigenous language*")) |
| AND (ALL(implementation OR "policy execution" OR "policy adoption" OR "policy enforcement"))) |
| OR (ALL("University of Cape Town") OR ALL("Stellenbosch University") OR ALL("University of KwaZulu-Natal") |
| OR ALL("University of Pretoria") OR ALL("University of the Witwatersrand") OR ALL("University of the Free State") |
| OR ALL("University of Johannesburg") OR ALL("University of the Western Cape") OR ALL("North-West University") |
| OR ALL("University of South Africa") OR ALL("Tshwane University of Technology") OR ALL("University of Fort Hare") |
| OR ALL("Rhodes University") OR ALL("University of Limpopo") OR ALL("Sefako Makgatho Health Sciences University") |
| OR ALL("University of Venda") OR ALL("Nelson Mandela University") OR ALL("Walter Sisulu University") |
| OR ALL("University of Zululand") OR ALL("Cape Peninsula University of Technology") OR ALL("Central University of Technology") |
| OR ALL("Durban University of Technology") OR ALL("Mangosuthu University of Technology") OR ALL("Sol Plaatje University") |
| OR ALL("University of Mpumalanga") OR ALL("Vaal University of Technology")) |
| AND (ALL("English") OR ALL("Afrikaans") OR ALL("isiXhosa") OR ALL("isiZulu") OR ALL("Sesotho") OR ALL("Sepedi") |
| OR ALL("Tshivenda") OR ALL("Xitsonga") OR ALL("Setswana") OR ALL("isiNdebele") OR ALL("siSwati")) |

and relevant languages spoken in South Africa, ensuring a holistic and inclusive approach. The inclusion of university names is intended to refine results to South African institutions, while the proximity operators and Boolean logic enhance the specificity and relevance of the search.

Google Scholar does not support proximity searching (NEAR/N) or complex Boolean logic, such as nested parentheses. Therefore, the search strategy has been simplified, as shown in Table 3. The search string includes key terms related to language policy and higher education in South Africa, incorporating Boolean operators (AND, OR) and truncation (*) where applicable. It also specifically includes the official South African languages to ensure relevance.

AJOL supports Boolean searches, but its functionality is more limited compared to databases like Scopus and Web of Science. Therefore, the search string has been tailored to focus on key terms relevant to language policy and higher education, while also considering AJOL's character limits and indexing constraints

**Planned limits.** The search will be restricted to literature published between 1994 and current, as this period captures key policy developments and reforms in language policy in South African higher education. A detailed rationale for this timeframe is provided in the Search Strategy section. Both peer-reviewed and grey literature will be included to ensure comprehensive coverage.

**Table 3. Search string 2 for google scholar (focused on South African language policy in higher education).**

| |
|---|
| ("language policy" OR "language planning" OR "language governance") |
| AND ("higher education" OR "university" OR "tertiary education" OR "postsecondary education" AND "South Africa") |
| AND (multilingual* OR bilingual* OR "language diversity" OR "indigenous language*") |
| AND (implementation OR "policy enforcement" OR "policy execution") |
| AND ("English" OR "Afrikaans" OR "isiXhosa" OR "isiZulu" OR "Sesotho" OR "Sepedi" OR "Tshivenda" |
| OR "Xitsonga" OR "Setswana" OR "isiNdebele" OR "siSwati" OR "Xitsonga") |

**Ethics statement.** This scoping review does not require formal ethical approval as it doesn't involve human participants or the collection of primary data. However, we will adhere to ethical research principles, including proper acknowledgment of sources, transparency in reporting, and maintaining academic integrity.

**Record management.** Search results managed using EndNote 21 for citation management and duplicate removal. Rayyan [19] will be used for title, abstract, and full-text screening.

**Study selection process.** Title and abstract screening: three reviewers will independently screen studies. Full-text review: eligibility confirmed based on predefined criteria.
Resolution of discrepancies: discussion or consultation with a fourth reviewer.

## Stage 3: Study selection

**Inclusion criteria.**

1. Studies that align with the PCC framework.

2. Focus on public HEIs in South Africa.

3. Address post-apartheid language policies and multilingual approaches.

**Exclusion criteria.**

1. Studies focused on private institutions or contexts outside South Africa (unless providing comparative insights).

2. Literature unrelated to post-apartheid language policy frameworks.

**Managing and documenting the screening process.** All studies passing the title screening stage will be managed using Rayyan [19] to ensure systematic organization throughout the abstract and full-text review stages. Rayyan will also be used to document decisions and reviewer notes consistently. In parallel, an Excel spreadsheet will be maintained to track the screening process, ensuring transparency and facilitating easy reference. The final results will be summarized and presented using a PRISMA flowchart, providing a clear overview of the selection process. For articles not freely available online, the University of KwaZulu-Natal library services will be utilized to access articles. If necessary, full texts will be requested directly from the authors to ensure comprehensive data inclusion.

## Stage 4: Data charting

A standardized data charting form (Table 5) will be created to ensure consistency in data collection. Three reviewers will independently extract data using this form. Any discrepancies will be resolved through discussion and consensus, with a fourth reviewer consulted if necessary.

**Table 4. Search string 3 for AJOL (focused on South African language policy in higher education).**

| |
|---|
| ("language policy" OR "language planning" OR "language governance") |
| AND (" higher education" OR "university" OR "tertiary education" OR "postsecondary education" AND "South Africa") |
| AND (multilingual* OR bilingual* OR "language diversity" OR "indigenous language*") |
| AND (implementation OR "policy enforcement" OR "policy execution") |
| AND ("English" OR "Afrikaans" OR "isiXhosa" OR "isiZulu" OR "Sesotho" OR "Sepedi" OR "Tshivenda" OR "Xitsonga" |
| OR "Setswana" OR "isiNdebele" OR "siSwati" OR "Xitsonga") |

**Table 5. Data charting form.**

| Category | Details |
|---|---|
| Study Characteristics | Author(s), year, university name, type of study, type of language policy, policy goals. |
| Implementation Details | Policy application specifics, challenges, strategies, and effectiveness. |
| Challenges and strategies | Identified challenges<br>Documented successes<br>Creative approaches to address barriers |
| Research design | Methodology, data collection methods, sample size, and analysis techniques. |
| Recommendations | Authors' conclusions, policy implications, and suggestions for further research. |

## Stage 5: Collating, summarizing, and analysing the results

Results will be presented using

1. Descriptive analysis: summarizing study characteristics using tables and charts.

2. Narrative synthesis: contextualizing qualitative findings within South Africa's socio-political landscape.

3. Thematic analysis: Identifying recurring themes and patterns in policy implementation.

If data homogeneity is sufficient, quantitative synthesis (e.g., meta-analysis) will be conducted. Otherwise, a narrative synthesis will be used.

### Additional analysis

Sensitivity and subgroup analyses will assess the robustness of findings, examining variations across institutional characteristics and studies of varying risk of bias. Meta-regression will be considered if sufficient data is available to explore associations between study characteristics, such as institutional size or policy age, and outcomes. These additional analyses will provide deeper insights into factors influencing policy success and improve the generalizability of findings.

### Synthesis approach

Following PRISMA-ScR guidelines, the analysis will begin with descriptive statistics to summarize the characteristics of the included studies, followed by a dual synthesis approach comprising narrative synthesis for qualitative insights and quantitative descriptive analysis for statistical trends. Results will be presented through narrative summaries contextualized within South Africa's socio-political and linguistic landscape, data visualizations such as implementation matrices, and summary tables.

### Key outcomes

Key outcomes will include the implementation of multilingual policies, challenges and barriers, successes and effective practices, and the impact on student outcomes such as academic performance and social integration. A thematic synthesis will identify patterns and themes in policy implementation, focusing on promoting multilingualism while addressing challenges and successes.

Quantitative synthesis, including meta-analysis, will be conducted if data homogeneity is achieved based on consistent outcome measures, study designs, and sufficient sample sizes. Effect sizes such as Cohen's d or odds ratios will be used, with heterogeneity assessed through the $I^2$ statistic, and random-effects models or subgroup analyses applied as needed. If data heterogeneity prevents meta-analysis, narrative synthesis will be employed instead.

### Risk of bias assessment

Risk of bias at the study level will be evaluated using frameworks such as ROBIS (18) or GRADE (19). High-risk studies will be weighted less or presented with caution.

### Evaluation frameworks

**Bias mitigation.** The strength of evidence will be evaluated using the GRADE system, ensuring reliable and transparent synthesis. Findings will culminate in actionable recommendations for policy and practice, considering institutional capacity and resource allocation, and will be disseminated through publications, conferences, and stakeholder engagement.

Risk of bias will be assessed at the study level using established frameworks such as ROBIS [20] or GRADE [21] to evaluate study reliability and validity. Studies with high risk of bias will be weighted less in the synthesis or presented with caution to ensure transparency. Publication bias will be evaluated using funnel plots or Egger's test, where applicable. Selective reporting will be examined by comparing the outcomes reported in studies to their protocols. These biases will be considered in data synthesis to maintain the accuracy and credibility of the findings.

### Stage 6: Stakeholder consultation

Key stakeholders (university administrators, policymakers, advocacy groups, students, and researchers) will be consulted to validate findings and gather feedback. Consultations will include focus groups, meetings, and surveys to ensure diverse perspectives are incorporated.

**Feedback collection.** Stage 6 will involve engaging key stakeholders to enhance collaboration and ensure the applicability of the findings for the development of effective language policies in South African higher education. Key stakeholders—such as university administrators, DHET policymakers, language advocacy groups, students, and academic researchers—will be consulted to provide valuable input on language policy implementation.

Meetings and focus groups will be used to present findings, facilitate discussions on policy implications, and gather feedback. This process will highlight areas of consensus and differing views on current practices. Structured feedback will be collected through surveys, open discussions, and written responses to draft reports, ensuring diverse perspectives are considered and documented.

**Data management.** Data will be managed using EndNote and Rayyan, with extraction and synthesis tracked in Excel. Final data will be made publicly available on OSF. Results will be disseminated through academic publications, reports, and stakeholder engagement sessions.

**Dissemination plan.** To ensure transparency and systematic management, reference management software like EndNote and Rayyan will be used. EndNote will organize and store records, while Rayyan will streamline the screening process and track reviewer decisions [22]. Data extraction will be conducted via Excel spreadsheets, with Rayyan used for managing reviewer notes. Thematic analysis will synthesize data on language policy implementation, with narrative synthesis and descriptive statistics presented in tables [17]. Stakeholder and reviewer feedback will be documented in Rayyan or separate Excel documents. To maintain data integrity and accessibility, all records will be stored on secure cloud platforms (e.g., Google Drive, Dropbox), with version control to track changes. The review data will be made publicly available on OSF for transparency. Final results will be shared through a PRISMA-ScR flowchart, summary tables, and visual aids in academic publications or reports.

## Discussion

Higher education institutions (HEIs) play a pivotal role in selecting mediums of instruction and fostering language profi-ciency to ensure equitable access to educational resources. While English remains the dominant language for publishing

and research, especially in graduate and postgraduate education, this predominance presents distinct challenges for non-native English speakers [23]. Studies have shown that non-native English-speaking scientists spend approximately 91% more time reading and 51% more time writing scientific papers compared to their native English-speaking counterparts [24]. Additionally, researchers whose first language is not English can spend around twice as long reading an English-language scientific journal article as native speakers[25].

To address these disparities, language policies in HEIs should support linguistic diversity alongside the use of English. Institutions must balance the necessity of English for global scientific communication with the promotion of other languages to foster inclusion and accessibility. This balance is particularly crucial given the challenges non-English speakers face in academic publishing and comprehension. By aligning language policies with their specific geographic and demographic contexts, HEIs can create more inclusive academic environments that recognize and mitigate the linguistic barriers faced by non-native English speakers.

## Acknowledgments

We extend our gratitude to the University of KwaZulu-Natal (UKZN) Library for providing access to the resources that supported the development of this protocol.

## Author contributions

**Conceptualization:** Silingene Joyce Ngcobo, Zamasomi Meyiwa Luvuno, Wilbroda Hlolisile Chiya, Celenkosini Thembelenkosini Nxumalo, Gugulethu Brightness Mazibuko, Busisiwe Purity Ncama, Sinegugu Evidence Duma, Deshini Naidoo.

**Methodology:** Tracy Zhandire, Zamasomi Meyiwa Luvuno, Wilbroda Hlolisile Chiya, Celenkosini Thembelenkosini Nxumalo, Gugulethu Brightness Mazibuko, Busisiwe Purity Ncama, Sinegugu Evidence Duma, Deshini Naidoo.

**Writing – original draft:** Tracy Zhandire, Silingene Joyce Ngcobo.

**Writing – review & editing:** Tracy Zhandire, Silingene Joyce Ngcobo, Zamasomi Meyiwa Luvuno, Wilbroda Hlolisile Chiya, Celenkosini Thembelenkosini Nxumalo, Gugulethu Brightness Mazibuko, Busisiwe Purity Ncama, Sinegugu Evidence Duma, Deshini Naidoo.

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
