## [Decision Letter · Decision Letter 0]

4 Feb 2025

PONE-D-24-54236Evaluating language policy implementation in South African higher education - three decades of progress and challenges: a scoping review protocolPLOS ONE

Dear Dr. Zhandire,

Thank you for submitting your manuscript to PLOS ONE. After careful consideration, we feel that it has merit but does not fully meet PLOS ONE’s publication criteria as it currently stands. Therefore, we invite you to submit a revised version of the manuscript that addresses the points raised during the review process.

The **search strategy**  requires greater breadth to ensure comprehensiveness, including the addition of **more synonyms**  for search terms, the inclusion of **specific university names**  and **languages** , and the use of **truncation (*) and proximity searching (NEAR/N)**  to refine the retrieval process. Additionally, the **language filter for English-only studies**  should be reconsidered, as it may contradict the study’s focus on multilingual policies. The **justification for restricting the search period to 1994–present**  should be clearly stated in the **Methods**  section. Furthermore, there is a need for **additional citations**  to support methodological choices, particularly for **Arksey & O’Malley’s framework, PRISMA-ScR guidelines, JBI Manual for Evidence Synthesis, the Population, Concept, Context framework, and Rayyan software** . Finally, the discussion should acknowledge the role of **English as the dominant language of scientific communication** , particularly in **graduate and postgraduate education** , where publishing requirements may present challenges for non-English speakers. Addressing these technical concerns will improve the study’s methodological rigor and alignment with best practices.

We look forward to receiving your revised manuscript.

Kind regards,

Muhammad Shahzad Aslam, Ph.D.,M.Phil., Pharm-D

Academic Editor

PLOS ONE

Journal requirements:   When submitting your revision, we need you to address these additional requirements. 1. Please ensure that your manuscript meets PLOS ONE's style requirements, including those for file naming. The PLOS ONE style templates can be found at https://journals.plos.org/plosone/s/file?id=wjVg/PLOSOne_formatting_sample_main_body.pdf and https://journals.plos.org/plosone/s/file?id=ba62/PLOSOne_formatting_sample_title_authors_affiliations.pdf. 2. Your abstract cannot contain citations. Please only include citations in the body text of the manuscript, and ensure that they remain in ascending numerical order on first mention.

Additional Editor Comments:

Justify restricting the search period to 1994–present in the Methods section.

• Enhance search strategy:

• Include more synonyms for search terms.

• Add specific university names and languages to be included in the study.

• Use truncation (*) and proximity searching (NEAR/N) for better search accuracy.

• Avoid filtering studies only in English, as this may contradict the focus on multilingual policies.

The paper should address how language barriers in science education particularly impact graduate and postgraduate students, given that English is the dominant scientific language.

Reviewers' comments:

Reviewer's Responses to Questions

**Comments to the Author**

1. Does the manuscript provide a valid rationale for the proposed study, with clearly identified and justified research questions?

Reviewer #1: Yes

Reviewer #2: Yes

Reviewer #3: Yes

Reviewer #4: Yes

Reviewer #5: No

2. Is the protocol technically sound and planned in a manner that will lead to a meaningful outcome and allow testing the stated hypotheses?

Reviewer #1: Yes

Reviewer #2: Yes

Reviewer #3: Yes

Reviewer #4: Yes

Reviewer #5: Partly

3. Is the methodology feasible and described in sufficient detail to allow the work to be replicable?

Reviewer #1: Yes

Reviewer #2: Yes

Reviewer #3: Yes

Reviewer #4: No

Reviewer #5: No

4. Have the authors described where all data underlying the findings will be made available when the study is complete?

Reviewer #1: No

Reviewer #2: Yes

Reviewer #3: No

Reviewer #4: Yes

Reviewer #5: Yes

5. Is the manuscript presented in an intelligible fashion and written in standard English?

Reviewer #1: Yes

Reviewer #2: Yes

Reviewer #3: Yes

Reviewer #4: Yes

Reviewer #5: Yes

6. Review Comments to the Author

You may also provide optional suggestions and comments to authors that they might find helpful in planning their study.

Reviewer #1: • Abstract: Please add the dates of coverage to the abstract ==methods section.

• The abstract exceeded 300 words. So, please revise it to follow the structured abstract recommended words in PLOS One Open’s Instructions for Authors for study submission. See: https://journals.plos.org/plosone/s/submission-guidelines#:~:text=The%20Abstract%20comes%20after%20the,objective(s)%20of%20the%20study

• Line 11: authors wrote specifically, it seeks to: So, they ended the preamble with a “:” and they went on to use a “,” to separate each specific objective. I suggest if a “:” is used, then a “;” is ideal to separate each point then end with a period. E.g., 1) map the integration of multilingual policies into institutional teaching, research, and administrative practices; (2) identify persistent barriers to effective policy implementation.

• Please, in the text, cite the reference number in square brackets e.g., “[19]” not in parenthesis “(19)”. Therefore, do thorough revision to effect this.

• Please, separate where two different articles are cited with “,” e.g., [10], [11] not (10) (11) remember it should be bracket square and not parenthesis.

• Wherever a “:” is used to indicate a list of points, please separate each point with a “;” and end with a period.

• Why is the search restricted to 1994 to present (three decades)? Please justify in the methods section.

• I suggest you delete the conclusion section for it is not required for protocol articles.

• Line 37: (3), argue for comprehensive language policies spanning all educational levels, including higher education, to enhance accessibility and inclusivity. I suggest reviewers can mention the author’s last name without the year of publication. E.g., Amoah [3], argues that for comprehensive language policies spanning all educational levels, including higher education, to enhance accessibility and inclusivity.

Reviewer #2: This was very clear, and even as a non-expert in research, I could follow the process. I find this proposal fascinating as a professor of education, because in the US, we often find working with 2-3 languages challenging! Your study is a good "state of the union" analysis to see how reforms are working and I look forward to reading what you discover. I only discovered a small error on page 3 line 37, where I think part of a sentence is missing.

Reviewer #3: This paper is written in the future tense. Have you done this study, or are you planning to do this study? I should think that PLOS One would want to publish the results of your study. I also would like to see more citations to your research tools - such as The review will follow Arksey and O'Malley's framework and PRISMA-ScR guidelines. And JBI Manual for Evidence Synthesis. We have some info under methodology/scoping review about Arksey….

Population, Concept, Context framework - ?? citation?

Rayyan software

With regard to science education, the global language of science currently is English – not so much a concern for undergraduate education but a concern for graduate and post-graduate education, where publishing is required.

Reviewer #4: The study proposed is sound and valuable to the research community. However, the search strategy, which is the core component of data collection, could greatly benefit from additional breadth to ensure comprehensiveness. More synonyms are needed for each search topic, as are included in the attached document. Specifically, authors should plan to include the names of the universities that could be included in the study if possible, as well as, the specific languages that would be included. Use of both truncation (*) and proximity searching (NEAR/N) should also be considered. For example, "Language" NEAR/2 (policy or policies). Proximity searching is available syntactically in Web of Science, Scopus, and ERIC.

Beyond the search, authors should consider eliminating the language filter for English described in the methodology. Although most studies are published in English, it seems counterproductive to exclude additional languages in a study about multilingual policies.

Reviewer #5: Dear authors,

although this scoping review protocol is quite well written, it is not suitable for publication in this journal, which accepts only systematic review and meta-analysis protocols.

7. PLOS authors have the option to publish the peer review history of their article (what does this mean? ). If published, this will include your full peer review and any attached files.

**Do you want your identity to be public for this peer review?** For information about this choice, including consent withdrawal, please see our Privacy Policy .

Reviewer #1: No

Reviewer #2: No

Reviewer #3: **Yes: ** Linda Billings

Reviewer #4: No

Reviewer #5: No

---

## [Author Response · Author response to Decision Letter 1]

21 Feb 2025

Subject: Resubmission of Revised Manuscript

Dear Editor

Thank you for your consideration of our manuscript and for providing us with the opportunity to submit a revised version. We appreciate the constructive feedback from the reviewers and the editorial team, which has helped us strengthen our work.

We have carefully addressed all the comments and suggestions provided, and we believe the revisions have enhanced the clarity and quality of our manuscript. Please find attached the revised version along with a detailed response to the reviewers' comments, outlining the changes made.

We sincerely appreciate your time and consideration and look forward to your further evaluation of our submission. Please let us know if any additional information is required.

---

## [Decision Letter · Decision Letter 1]

19 Mar 2025

Evaluating language policy implementation in South African higher education - three decades of progress and challenges: a scoping review protocol

PONE-D-24-54236R1

Dear Dr. Zhandire,

We’re pleased to inform you that your manuscript has been judged scientifically suitable for publication and will be formally accepted for publication once it meets all outstanding technical requirements.

Kind regards,

Muhammad Shahzad Aslam, Ph.D.,M.Phil., Pharm-D

Academic Editor

PLOS ONE

Additional Editor Comments (optional):

Reviewers' comments:

Reviewer's Responses to Questions

**Comments to the Author**

1. Does the manuscript provide a valid rationale for the proposed study, with clearly identified and justified research questions?

Reviewer #4: Yes

2. Is the protocol technically sound and planned in a manner that will lead to a meaningful outcome and allow testing the stated hypotheses?

Reviewer #4: Yes

3. Is the methodology feasible and described in sufficient detail to allow the work to be replicable?

Reviewer #4: Yes

4. Have the authors described where all data underlying the findings will be made available when the study is complete?

Reviewer #4: Yes

5. Is the manuscript presented in an intelligible fashion and written in standard English?

Reviewer #4: Yes

6. Review Comments to the Author

You may also provide optional suggestions and comments to authors that they might find helpful in planning their study.

Reviewer #4: Authors have responded to comments and made appropriate changes to the protocol. The search is greatly improved.

7. PLOS authors have the option to publish the peer review history of their article (what does this mean? ). If published, this will include your full peer review and any attached files.

**Do you want your identity to be public for this peer review?** For information about this choice, including consent withdrawal, please see our Privacy Policy .

Reviewer #4: No

---

## [Editor Report · Acceptance letter]

PONE-D-24-54236R1

PLOS ONE

Dear Dr. Zhandire,

I'm pleased to inform you that your manuscript has been deemed suitable for publication in PLOS ONE. Congratulations! Your manuscript is now being handed over to our production team.

Kind regards,

on behalf of

Dr. PLOS Manuscript Reassignment

Staff Editor

PLOS ONE